# Spatiotemporal analysis of medical resource deficiencies in the U.S. under COVID-19 pandemic

**Dexuan Sha**[1,2]**, Xin Miao**[3]**\*, Hai Lan**[1,4]**, Kathleen Stewart**[4]**, Shiyang Ruan**[2]**, Yifei Tian**[1]**,
**Yuyang Tian**[5]**, Chaowei Yang**[1,2]

**1** NSF Spatiotemporal Innovation Center, George Mason University, Fairfax, VA, United States of America,
**2** Department of Geography and GeoInformation Science, George Mason University, Fairfax, VA, United
States of America, **3** Department of Geography, Geology and Planning, Missouri State University,
Springfield, MO, United States of America, **4** Department of Geographical Sciences, University of Maryland,
College Park, MD, United States of America, **5** Mercy Clinic Family Medicine, Springfield, MO, United States
of America

\* xinxiao@missouristate.edu

pone.0240348

UNITED STATES

**Data Availability Statement:** All relevant data are
available from GitHub (https://github.com/
stccenter/COVID-19-Data/tree/master/US).

## Abstract

Coronavirus disease 2019 (COVID-19) was first identified in December 2019 in Wuhan,
China as an infectious disease, and has quickly resulted in an ongoing pandemic. A data-
driven approach was developed to estimate medical resource deficiencies due to medical
burdens at county level during the COVID-19 pandemic. The study duration was mainly
from February 15, 2020 to May 1, 2020 in the U.S. Multiple data sources were used to
extract local population, hospital beds, critical care staff, COVID-19 confirmed case num-
bers, and hospitalization data at county level. We estimated the average length of stay from
hospitalization data at state level, and calculated the hospitalized rate at both state and
county level. Then, we developed two medical resource deficiency indices that measured
the local medical burden based on the number of accumulated active confirmed cases nor-
malized by local maximum potential medical resources, and the number of hospitalized
patients that can be supported per ICU bed per critical care staff, respectively. Data on med-
ical resources, and the two medical resource deficiency indices are illustrated in a dynamic
spatiotemporal visualization platform based on ArcGIS Pro Dashboards. Our results pro-
vided new insights into the U.S. pandemic preparedness and local dynamics relating to
medical burdens in response to the COVID-19 pandemic.

## 1. Introduction

Coronavirus disease 2019 (COVID-19) was first identified in December 2019 in Wuhan,
China as an infectious disease, and has quickly resulted in an ongoing pandemic. Just before
the global pandemic COVID-19, a report by the Global Health Security Index was released,
which is the first-ever comprehensive ranking of 195 countries based on their pandemic pre-
paredness, with six categories of 140 questions and 34 indicators [1]. Although national health
security is fundamentally weak across the globe, the U.S. scored 83.5/100 and ranked No.1 in

**Funding:** X.M. acknowledges support from NSF CSSI-1835512. C.Y. is supported by NSF CNS-1841520 and CSSI-1835507. (www.nsf.gov/) The funders had no role in study design, data collection and analysis, decision to publish, or preparation of the manuscript.

**Competing interests:** The authors have declared that no competing interests exist.

the report. As evidence, there were 34.7 critical care beds per 100,000 inhabitants in the U.S. by 2009, which is higher than that of any other country [2, 3]. However, the U.S. has fewer hospital beds (2.8), and practicing physicians (2.6) per 1,000 capita compared to other similar large and wealthy countries [4].

Since the COVID-19 outbreak, it has been estimated that a significant percentage of the U.S. population would test positive for COVID-19 even given a conservative estimation [5]. For example, a recent AHA (American Hospital Association) webinar on COVID-19 projected that 30% (96 million) of the U.S. population would test positive, with 5% (4.8 million) being hospitalized, 2% (1.9 million) would be admitted to the intensive care unit (ICU), and 1% (960,000) would require ventilators [6]. This projection is generally compatible with the characteristics of COVID-19 in Wuhan, China, where 5% of patients required the intensive care unit and 2.3% required a ventilator [7]. Based on a recent CDC survey, the actual weekly hospitalization rate in April 2020 was around 5.8–7.5% for 100 counties across 14 states [8], which means a large number of infected patients will swarm into hospitals and ICUs. As a matter of fact, the U.S. had the highest number of confirmed cases of COVID-19 (82,404) in the world on March 26, 2020, and surpassed Italy for the highest national death toll (20,413) on April 11, 2020 [9, 10].

Are U.S. medical resources enough to handle the worst scenario during this crisis? The Society of Critical Care Medicine (SCCM) released a report regarding the medical resources both available and needed for a potentially overwhelming number of critically ill patients [6]. In this report, three fundamental elements or features, i.e. ventilators, ICU beds, and critical care staff (CCS) were identified as medical resources to plan for or manage a COVID-19 pandemic, and it would be wise to consider the interconnections among these factors in a spatiotemporal data analysis framework. Specifically, the medical resource distribution should be correlated with COVID-19 pandemic statistics in space (2D) and time (1D). So medical resource burden or deficiency can be identified through feature selection, visualization, monitoring, and cluster analysis [11].

Among the three elements mentioned above, an inventory of ventilators is difficult to quantify for estimating critical supply shortages. Based on a 2009 AHA survey, a total of 5,752 U.S. acute care hospitals were estimated to have 62,188 full-featured mechanical ventilators and 98,738 ventilators with limited features [12]. The Strategic National Stockpile (SNS) had an estimated 8,900 ventilators for emergency deployment in 2010, and between 12,000 and 13,000 ventilators by March 13, 2020 [13–15]. Based on these numbers, the ventilator inventory was approximately 173,000–174,000 in the U.S. A model-based analysis suggested that US hospitals could absorb between 26,200 to 56,300 additional ventilators at the peak of a national pandemic with robust pre-pandemic planning [16]. Since SNS can deliver ventilators within 24–36 hours after being requested by states and approved by federal organizations, and no reliable database for ventilator inventory exists at county or state level, we will not consider this factor in our spatiotemporal analysis. A recent model-driven study simply assumes one ventilator per critical care bed [17] and we use this same assumption in our analysis.

Hospital beds, especially ICU beds, are an important factor in evaluating medical resource deficiency during the COVID-19 pandemic, and quantity of beds has been used as a major factor in model-driven predictions of local critical care capacity limit [17, 18]. However, safe use of ventilators in ICU requires trained personnel. In a previous study, the number of trained medical personnel is assumed to correlate with the number of staffed beds maintained by hospitals [16]. This assumption is perhaps unrealistic at county level without considering the geographic disparity.

For this research, we assumed that a realistic measurement of the medical burden at county level should consider both ICU beds and critical care staff (CCS), which will provide

reasonable evidence for stakeholder (e.g., hospital, county and State governments policy and decision-making). In this study, we (1) conduct a medical data analysis, and re-evaluate the spatial distribution of medical resource features (hospital beds, ICU beds, and CCS) at county level; (2) develop two Medical Resource Deficiency Indices (MRDI and $MRDI_d$) by linking positive COVID-19 infections and local medical resources to measure local medical burden; and (3) develop a data-driven dynamic spatiotemporal framework to visualize and analyze the MRDI /$MRDI_d$ trends at the county level. Our results provided a new dimension of insight into the U.S. pandemic preparedness and local dynamic medical burden during COVID-19 pandemic. The dataset is open sourced and hosted on GitHub (https://github.com/stccenter/COVID-19-Data/tree/master/US), and are visualized through ArcGIS Dashboards at: http://mrd-dashboard.stcenter.net/.

## 2. Data

### 2.1. Base map and unit of analysis

A total of 3,143 counties and county-equivalents in the U.S. are used as the primary unit of this study, since they are manageable in a GIS system and small enough to reflect local geographic discrepancies. The base map was downloaded from the 2019 TIGER/LINE products from the U.S. Census Bureau, which is the most comprehensive spatial dataset designed for GIS platforms [19]. The county vector layer delineates the administrative boundary with land/water area without any demographic data, but it provides geographic entity codes (GEOIDs) for joining with other socio-economic data such as Census data. Based on the attributes of our collected medical-related datasets, we also prepared state and ZIP code boundaries for data fusion and integration at county level.

### 2.2. Medical resource feature extraction

In this study, two fundamental features of medical resources in the U.S. were extracted, i.e., hospital beds and critical care staff. Besides, the population and 60+ senior population data was extracted at county level from KHN online database [20], which is used to normalize the local medical data in the subsequent analysis.

 **2.2.1. Hospital beds.** National public and private online datasets were used to prepare county-level hospital bed counts. Hospital data were collected from Definitive Healthcare [21]. Definitive Healthcare consulting services share their hospital dataset to the entire health research community through ArcGIS online, which cover information of nationwide bed capacity and average yearly bed utilization of hospitals. Although it is not a real-time dataset that reflects each hospital's bed capacity during COVID-19, it can be used as a baseline to estimate the geographic disparity of local health resources.

 A hospital is defined as a healthcare institution providing inpatient, therapeutic, or rehabilitation services under the supervision of physicians with the capability of inpatient care [21]. All types of hospitals are included in our study. Five types of hospital beds are clearly identified in the Definitive Healthcare dataset. In our study, two hospital bed capacities were selected and used in the analysis. The first one is the number of licensed beds, which is the potential or maximum number of beds for which a hospital holds a license to operate. The second type of capacity refers to the number of adult ICU beds that could be used for COVID-19. During this crisis, hospitals could use additional intensive care beds to supplement an influx of patients. Therefore, adult ICU beds include not only internal medical ICU beds, but also burn, surgical, and trauma ICU beds. However, pediatric, premature or neonatal ICU beds are not included because they are mainly for a different target patient population, which has a much lower incidence rate of COVID-19.

Two other independent data sources of hospital beds are compared with the data from Definitive Healthcare. One is from Kaiser Health News (KHN) based on reports of ICU beds in 2018–2019 [20], and the other is from Homeland Infrastructure Foundation-Level Data (HIFLD) for licensed hospital beds updated on October 7, 2019 [22]. We conducted a regression analysis comparing KHN with Definitive Healthcare in terms of ICU beds, and comparing HIFLD with Definitive Healthcare in terms of licensed beds, and the coefficients of determination ($r^2$) are 0.94 and 0.97, respectively. The results validated the quality of the Definitive Healthcare dataset.

**2.2.2. Critical care staff.** A dataset of critical care staff (CCS) was extracted from the weekly updated National Provider Identifier Registry (NPI) database (~7.1 GB) through structured query language (SQL) [23]. The NPI is a unique 10-digit identification number for each health-care provider issued by the Centers for Medicare Medicaid Services through the National Plan and Provider Enumeration System. Each health-care provider could have multiple taxonomy codes, which indicate areas of specialization. Through consulting with medical researchers and front-line physicians, we extracted detailed CCS data from the NPI database released on April 15, 2020 as a medical resource feature (Table 1). Our study identifies 197,061 health care providers by searching unique NPI records and removing duplicate records. With the development of COVID-19 in the U.S., all these ICU-related staff (emergency medicine physician, critical care physicians, anesthesiologists, hospitalists, pulmonologist, infectious disease physician, surgery, anesthesiologist assistant, critical care nurses, nurse anesthetist, and respiratory therapists trained in mechanical ventilation) would become valuable but limited asset for critically ill ventilated patients [6].

## 2.3. COVID-19 patients

The U.S. Centers for Disease Control and Prevention (CDC) published daily COVID-19 confirmed cases on February 25, 2020. Each state got involved soon after and began to report COVID-19 data, including the daily and accumulated test and confirmed case numbers, hospitalization data, and death numbers at state level. However, numbers of discharged or released patients from hospitals are less widely available, e.g., only a few states, such as Maryland, Colorado, and New York provide some (incomplete) statistics on recovered patients from both hospital and home. This study mainly uses the data collected by the NSF Spatiotemporal Innovation Center (STC) at George Mason University. This dataset uses a datacube structure for

**Table 1. Critical care staff extracted from NPI database.**

| | Critical Care Staff (CCS) | Taxonomy Code | Number | Total Number* |
|---|---|---|---|---|
| Physician | Emergency Medicine | 207P00000X | 67591 | 131519 |
| | Anesthesiology (Critical Care Medicine) | 207LC0200X | 1871 | |
| | Hospitalist | 208M00000X | 27827 | |
| | Internal Medicine (Infectious Disease) | 207RI0200X | 11299 | |
| | Internal Medicine (Critical Care Medicine) | 207RC0200X | 10976 | |
| | Internal Medicine (Pulmonary Disease) | 207RP1001X | 19990 | |
| | Surgery (Surgical Critical Care) | 2086S0102X | 2392 | |
| Physician Assistant | Anesthesiologist Assistant | 367H00000X | 2953 | 2953 |
| Nurse | Certified & registered Nurse Anesthetist | 367500000X | 61585 | 62589 |
| | Nurse Practitioner (Critical Care Medicine) | 363LC0200X | 1040 | |
| Technician | Certified Respiratory Therapist | 2278C0205X | 164 | 538 |
| | Registered Respiratory Therapist | 2279C0205X | 379 | |
| Total | - | - | - | 197,061 |

* Duplicate records were removed since one health care provider may have multiple Taxonomy Codes.

spatiotemporal data aggregation from multiple sources. The data is cleaned, standardized, and updated daily to solve any data conflicts, and a time-series summary at state and county level is provided for the U.S. [10, 24].

The numbers of county-level confirmed positive cases as well as deaths were originally extracted from USA Facts based on CDC data [25], and compared with local public health agencies for verification. The confirmed and death cases reflect cumulative statistics since January 22, 2020, the day after the first confirmed cases were reported in Washington State. Furthermore, state level test and hospitalization data were extracted from the COVID Tracking Project [26]. However, the current and accumulated hospitalization cases from state health departments are largely incomplete. By April 29, 2020, a total of 22 states reported both current and accumulated hospitalized patient numbers, 17 states reported only current hospitalized numbers, and 10 states only reported accumulated hospitalized numbers, while Washington, D.C., Nevada and Nebraska did not provide information on the number of hospitalized cases.

## 3. Methods

Our analysis was mainly based on the publicly available data of the new confirmed daily cases reported for the U.S. from the 25th of February until the 1st of May, 2020. All data were fully anonymized.

### 3.1. Medical feature extraction and aggregation

Raw datasets in this study were collected from multiple sources with heterogeneous formats and structures. All data were processed and aggregated at county level based on County Federal Information Processing Standard (FIPS). Several aggregation methods were used for each raw dataset, as summarized in Fig 1.

First, the hospital data was originally presented as a point location in a coordinate format, and its attribute table includes five types of hospital beds. The spatial point aggregation algorithm was used to integrate the numbers of licensed beds and adult ICU beds at county level. The bed numbers per 1,000 residents were also calculated at county level.

The primary practice addresses of CCS were imported from the NPI database, and 5-digital ZIP codes were extracted. The total number of CCS within a county was counted based on the county's ZIP codes through geocoding and the point/ polygon aggregation algorithm. The number of CCS per 1,000 residents were also calculated at county level.

The accumulated COVID-19 confirmed case numbers were extracted at county level. We used existing hospitalization data to estimate the average length of stay (ALOS) in acute care, since it is key for estimating the daily hospitalized patients. For a given state, the current hospitalized patients should be equal to the accumulation of hospitalized patients minus the accumulation of deaths and discharged patients within the most recent ALOS. Since no patient discharge data was available, we assumed that the number of discharged patients was zero. Therefore, we estimated ALOS by matching (1) the accumulation of hospitalized patients minus the accumulation of deaths in most recent days, and (2) the current number of hospitalized patients, and finally interpolating by two nearest days or accumulation periods. It turns out to be an optimization problem to find a parameter (n) to match the two data sources, as shown in Eq (1).

$$\mathrm{ALOS} = \arg\min_{n \in \Re^+}(N_{h,n} - N_{death,n}) - N_{ch} \tag{1}$$

where $N_{h,n}$ is the accumulated number of hospitalized patients in the past n days, $N_{death,n}$ is the accumulated number of deaths in the past n days, and $N_{ch}$ is the number of currently hospitalized patients.

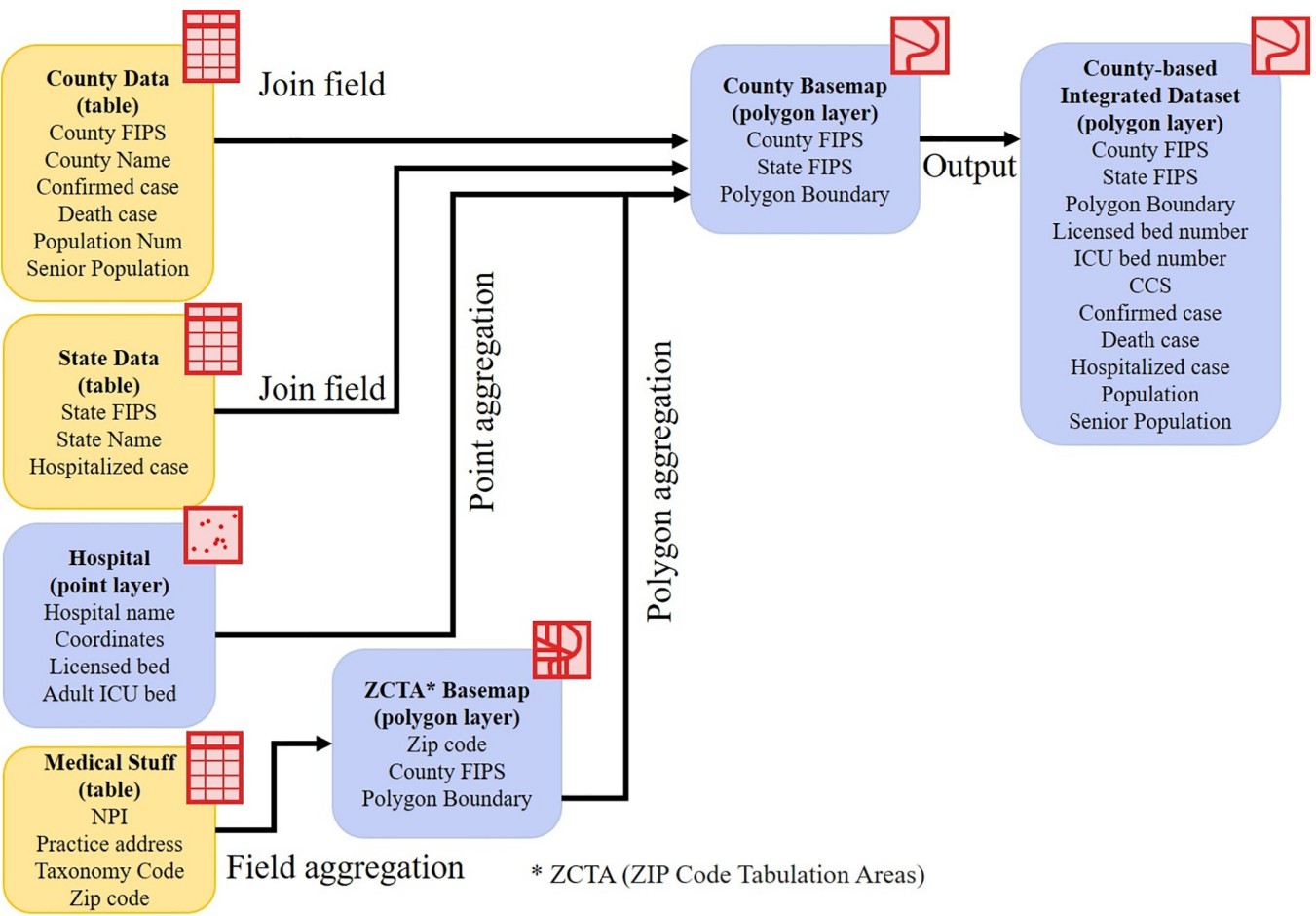

**Fig 1. Medical feature extraction workflow.**

State hospitalization data were only available recently (starting from March 17, 2020 in NY) with numerous missing data. By May 1, 2020, among 22 states that have both current and accumulated numbers of hospitalized patients, eight states (Colorado, Massachusetts, Maine, Minnesota, Montana, North Dakota, New York, Oklahoma) had complete data for the most recent 20 days; 12 states (Oklahoma, Wisconsin, Mississippi, Maryland, New Hampshire, New Mexico, Oregon, South Dakota, Virginia, Wyoming, Rhode Island, Kentucky) only had data in the most recent 5–15 days; and data from Arkansas, Arizona, and Connecticut were abandoned due to poor quality. We calculated the daily ALOS for these 19 states and pooled the results in Fig 2. The state ALOS ranges from 8.8 (New Mexico)-28.5 (Mississippi) days. The overall national ALOS weighted by state hospitalized patients is 15.5 days, which is longer than a previous estimation that the ALOS in acute care were 11 days [18]. It is worth noting that ALOS is likely to be underestimated since we assumed no discharged patients. Furthermore, ALOS is subject to change when more hospitalization data become available in the future.

Finally, we define the COVID-19 *hospitalized rate* as the ratio of the number of current hospitalized patients and the accumulated confirmed case numbers during the most recent ALOS. If the hospitalized rate remains the same within a state, the daily hospitalized patient number in a county can be estimated by using the accumulated COVID-19 confirmed case numbers minus deaths in the most recent ALOS, multiplied by the state average hospitalized rate. If no

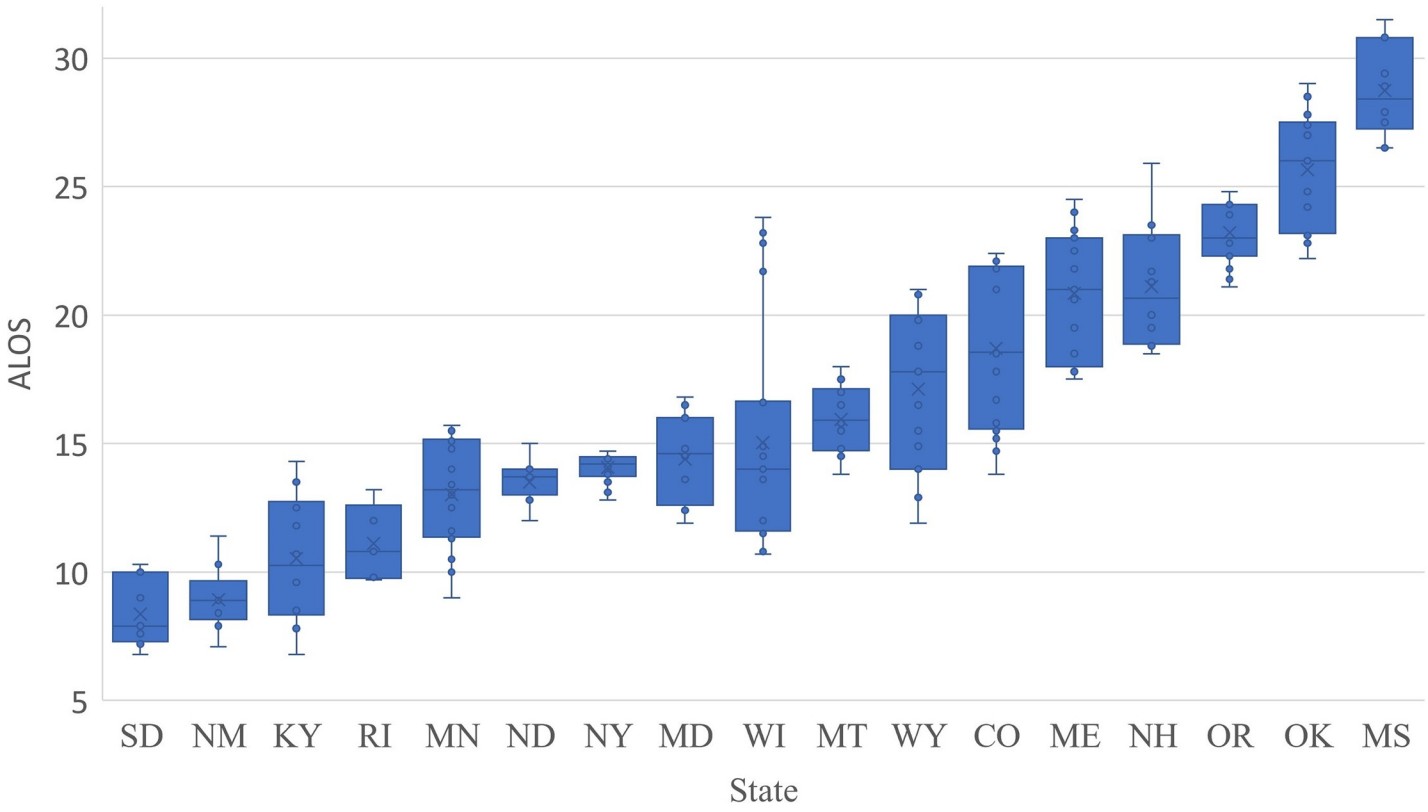

**Fig 2. Box-plot (5-number summary) of hospitalized ALOS among 19 states.**

state ALOS is available, we use the overall national average ALOS of 15.5 days. This daily hospitalized patient number can be used to evaluate the daily medical burden at county level.

## 3.2. Medical resource deficiency indices

The medical resource deficiency indices (MRDI) are defined as an indicator of medical resource burden at county level. We define two forms of MRDI: general MRDI, and local daily MRDI (MRDI_d).

$$\text{MRDI} = \frac{N_c - N_{death}}{N_{licbed} \cdot N_{CCS}} \tag{2}$$

where $N_c$ is the accumulated number of confirmed COVID patients, $N_{death}$ is the accumulated number of deaths, $N_{licbed}$ is the total number of licensed beds, and $N_{CCS}$ is the number of critical care staff. We assumed that $N_{licbed}$ and $N_{CCS}$ were relatively independent at county level, and the product of them represents the interconnection of these two medical resource features or factors. Therefore, the MRDI represents the number of accumulated active confirmed cases normalized by the local maximum potential medical resources (total licensed beds and total CCS). MRDI_d is represented as

$$\text{MRDI}_d = \frac{(N_{cA} - N_{dA}) \cdot r_h}{N_{icubed} \cdot N_{CCS}} \tag{3}$$

where $N_{cA}$ is the accumulated confirmed case numbers during a most recent ALOS, $N_{dA}$ is the accumulated death numbers during the same ALOS, $r_h$ is the state hospitalized rate derived

from state hospitalization data, and $N_{icubed}$ is the number of adult ICU beds. $MRDI_d$ represents the local daily medical burden, or the number of hospitalized patients that can be supported per ICU beds per CCS. $MRDI_d$ is large (>1) when local medical resources cannot fully support the hospitalized critically ill patients, or the local medical burden is heavy; and $MRDI_d$ is small (<1) when local medical resources are sufficient.

## 3.3. Visualization analysis using ArcGIS Dashboards

Based on ArcGIS Dashboard, we designed a comprehensive operational dashboard for monitoring, analyzing, visualizing, and sharing our medical data and analyzed results. A multi-stacked map is built at the center of the interface (Fig 3), which represents the spatial distributions of COVID-related statistics such as MRDI, death rate, and infection rate at county level over the U.S. In addition to visualizing the macro spatial distribution pattern of those statistics results, two lists of counties are displayed. Those counties are dynamically filtered by the current map extent in map view and are ranked in real-time by hospitalized rate and death rate to represent the spreading of COVID-19 and the outbreak situation in the selected study area. Focusing on a specific county, an indicator and two pie charts are applied to display for each county (Fig 4): 1) the comparison of active COVID—19 cases and the number of overall beds; 2) the percentage of ICU beds in overall beds; and 3) the proportion of each type of CCS. From the temporal analysis perspective, a time series chart is designed to demonstrate the dynamics of medical resource deficiencies for each county on a daily basis during the pandemic. In the following section, we will use the dashboard components to analyze spatiotemporal distributions of medical resource deficiencies. We will further explore the possible factors relating to the medical resource deficiencies for specific counties and areas as well as the medical resource capacity for non-severe COVID-19 patients, the supplies needed for severe cases, and proportion of each type of CCS.

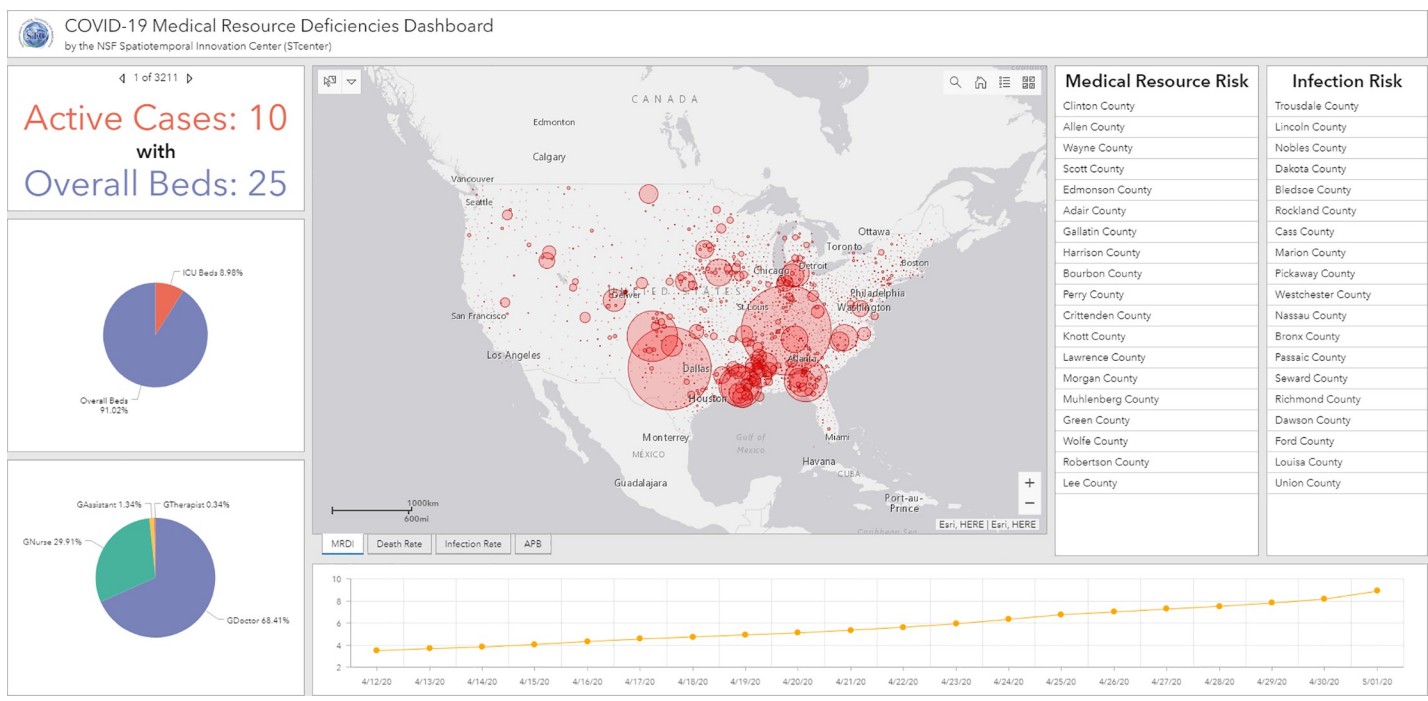

**Fig 3. Spatiotemporal visualization interface based on ArcGIS Dashboards.**

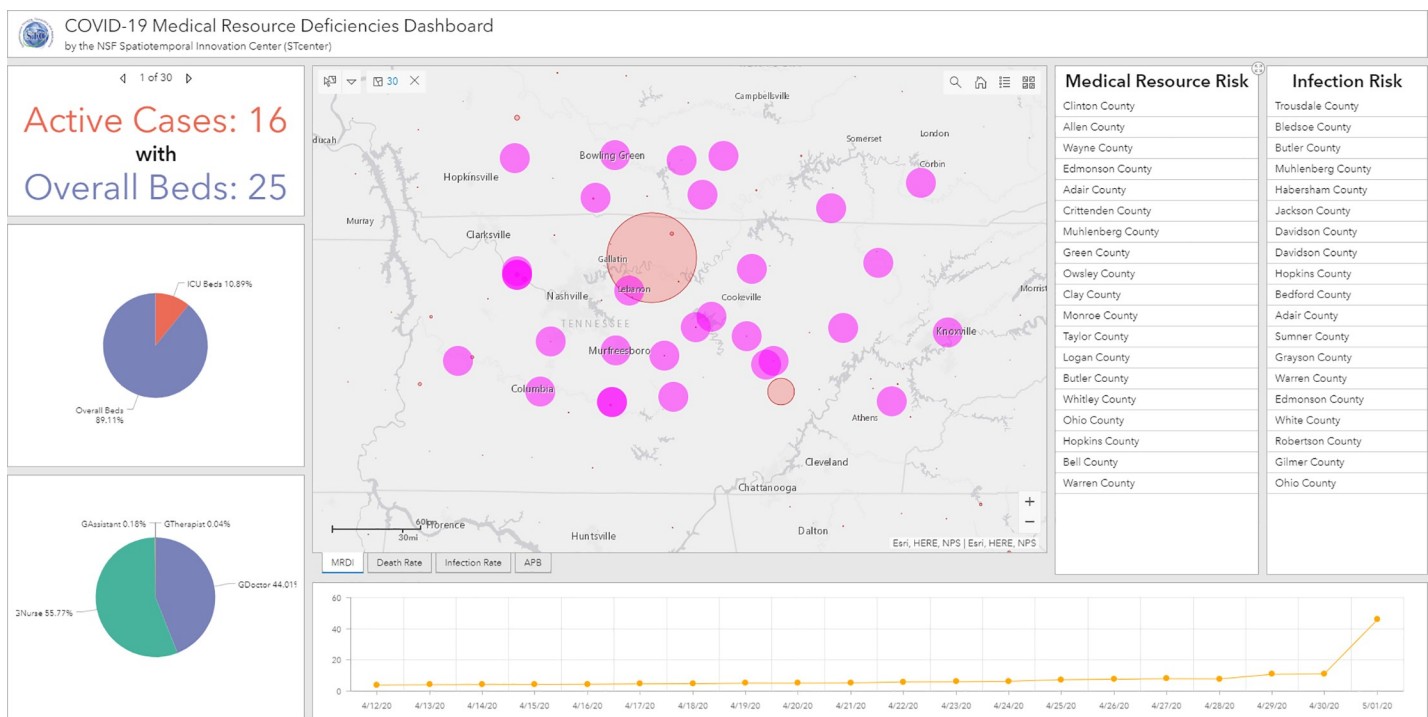

**Fig 4. A use case for regional visual analysis of Tennessee.**

## 4. Results

### 4.1. Medical resource features

The ICU beds per 1,000 residents (Fig 5A) and CCS per 1,000 residents (Fig 5B) are mapped at county level. Both maps show that these two medical resources are not homogeneously distributed across the U.S. Some midwestern states, such as North Dakota, South Dakota, Nebraska, Kansas, and Montana have more ICU beds, but less CCS. The spatial distribution of CCS shows a checker board pattern, with many gaps or low numbers across the country. The product of ICU beds and CCS per 1,000 residents is shown in Fig 6A. The darkest green zones represent counties with higher quantities of medical resources including ICU beds and CCS.

A total of 19 major medical centers represent top ranking healthcare facilities in the U.S. (Table 2) [27]. Medical centers are conglomerations of health care facilities including hospitals and research facilities that could be affiliated with a medical school. Overlaying the locations of these 19 medical centers on the map (purple circles on the map), it seems these counties and medical centers are spatially highly correlated (Fig 6A).

Since senior people (aged 60+) are vulnerable to COVID-19, we also produced a map of the product of ICU beds and CCS per 1,000 senior residents (Fig 6B). This map represents locations where the supply of medical resources for seniors is higher.

A regression analysis was conducted to examine the correlation between CCS and adult ICU beds at county level (Fig 7). If all 3,143 counties are included, the coefficient of determination ($r^2$) is 0.90. However, this high $r^2$ value is quite misleading, since it is heavily influenced by several large counties with rich medical resources (blue dots). Removing the top 30 counties, causes the coefficient of determination ($r^2$) to drop to 0.78, which better represents the geographic disparity of these two factors in most (3113) of the U.S. counties, as shown in Fig 5A and 5B.

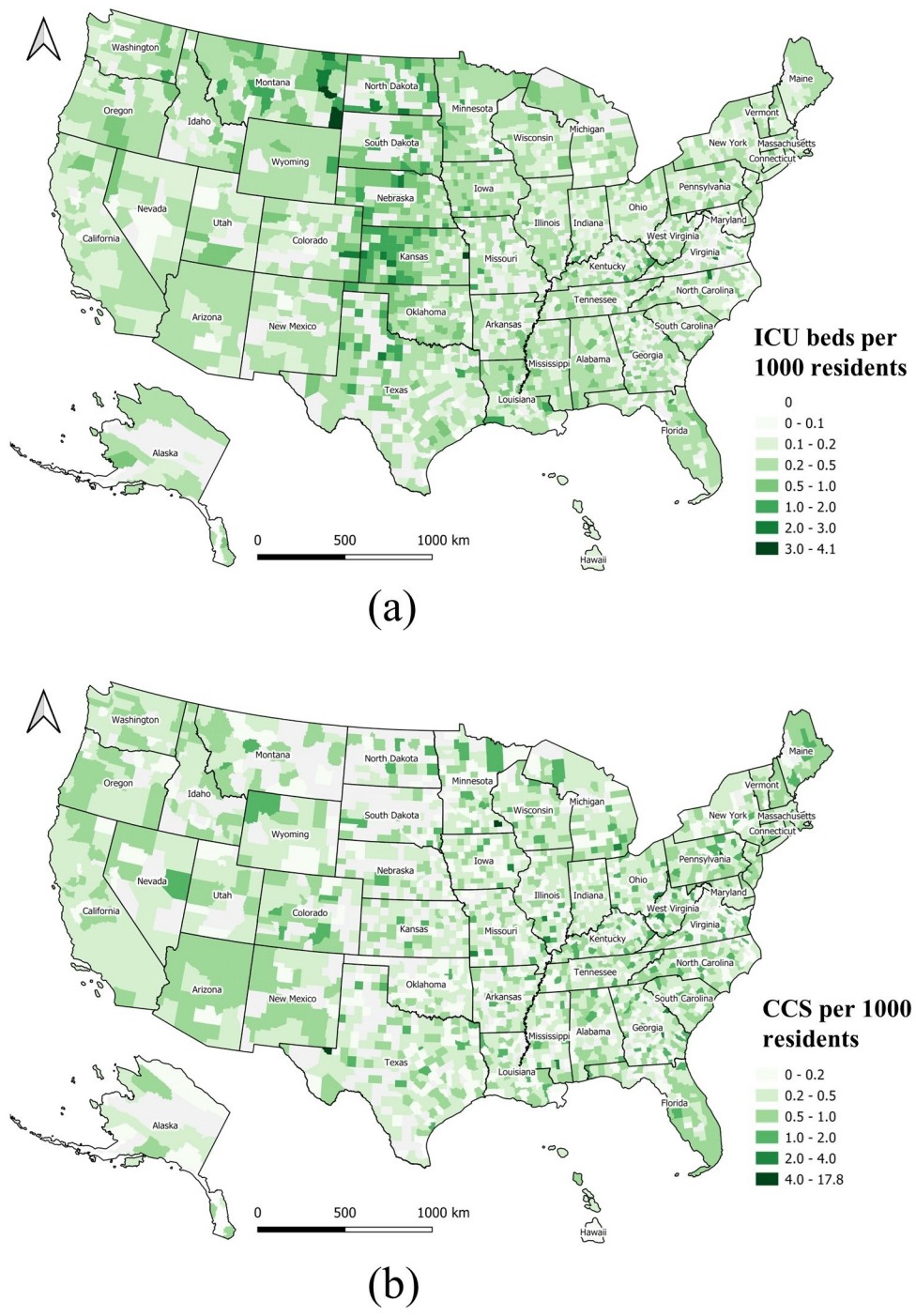

**Fig 5. Geographical distribution of medical resources at county level normalized by local population.** (a) ICU beds per 1,000 residents; (b) CCS per 1,000 residents.

A total of 671 counties have neither ICU beds nor CCS, and are shown in Fig 8. These counties are mainly distributed in less-populated rural areas across the U.S., and they are not included in MRDI or MRDI$_d$ calculation to avoid a divide-by-zero error. During the COVID-19 pandemic, individuals requiring a higher level of care in these areas would be sent to

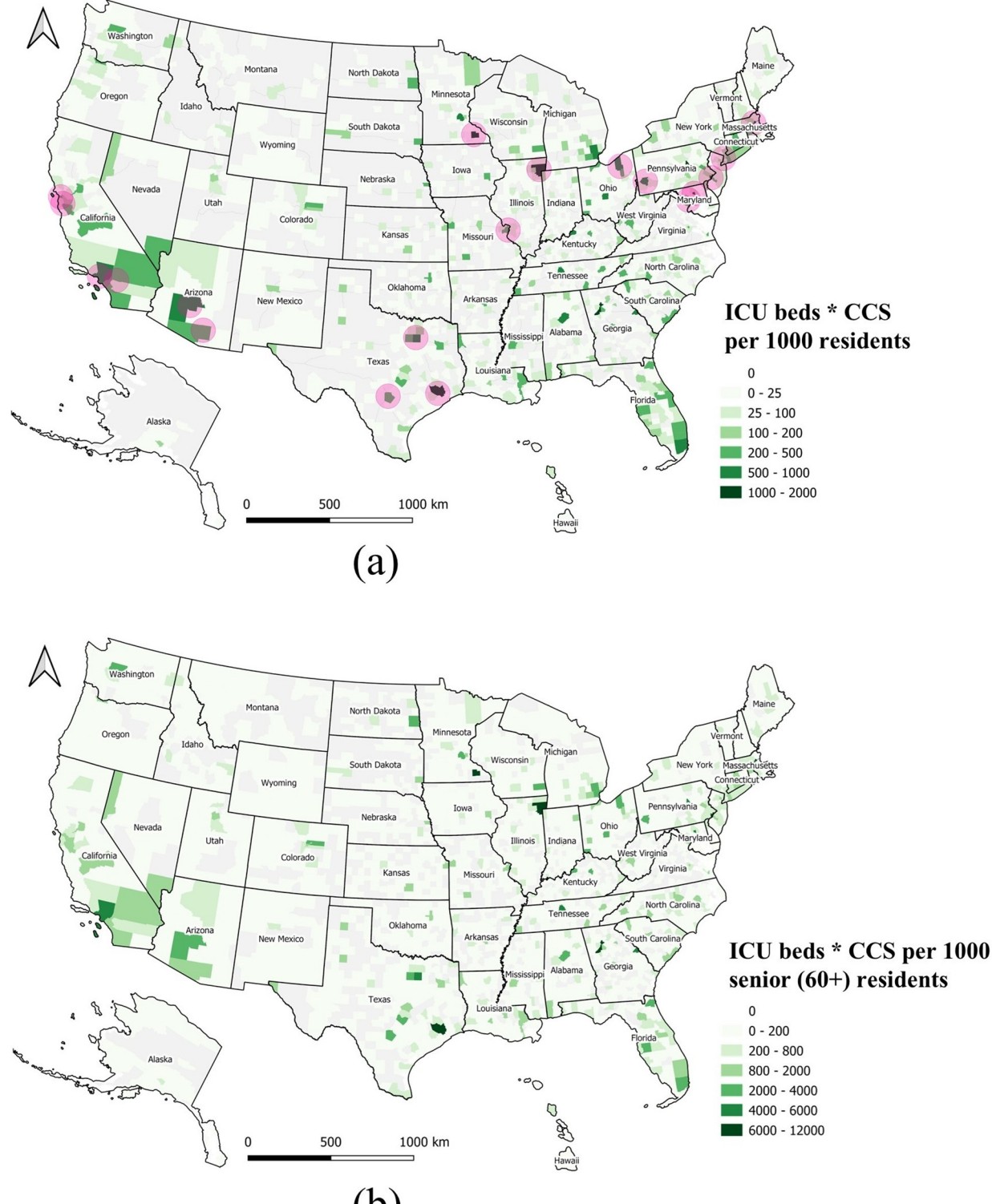

**Fig 6.** Overall medical resources at county level normalized by local population (a) The product of ICU beds and CCS per 1,000 residents, and 19 medical centers shown as purple bubbles; (b) The product of ICU beds and CCS per 1,000 for senior residents (aged 60+).

**Table 2. Major medical centers in the U.S.**

| # | Medical Center | Location |
|---|---|---|
| 1 | Banner University Medical Center Tucson | Tucson, AR |
| 2 | Phoenix Healthcare Cluster | Phoenix, AR |
| 3 | Loma Linda University Medical Center | Loma Linda, CA |
| 4 | Stanford University Medical Center | Stanford, CA |
| 5 | UCSF Medical Center | San Francisco, CA |
| 6 | Ronald Reagan UCLA Medical Center | Los Angeles, CA |
| 7 | Illinois Medical District | Chicago, IL |
| 8 | National Institutes of Health | Bethesda, MD |
| 9 | Johns Hopkins | Baltimore, MD |
| 10 | Boston Longwood Area | Boston, MA |
| 11 | Washington University Medical Center | St. Louis, MO |
| 12 | Mayo Clinic | Rochester, MN & others |
| 13 | NewYork-Presbyterian Hospital | New York, NY |
| 14 | Cleveland Clinic | Cleveland, Oh |
| 15 | University of Pennsylvania Health System | Philadelphia, PA |
| 16 | University of Pittsburgh Medical Center | Pittsburgh, PA |
| 17 | Texas Medical Center | Houston, TX |
| 18 | South Texas Medical Center | San Antonio, TX |
| 19 | Southwestern Medical District | Dallas, TX |

neighboring counties with sufficient medical resources, and could result in larger $MRDI_d$ in the neighboring counties.

## 4.2. Spatiotemporal trend of MDRI and MDRI$_d$

The spatiotemporal dynamics of general MDRI across the U.S. is illustrated at: http://mrd-dashboard.stcenter.net/. The general MDRI represents the number of accumulated active confirmed COVID-19 cases normalized by local maximum potential medical resources, while the dynamic view provides an insightful alternative visualization of COVID-19 U.S. cases by county. Six snapshot maps are illustrated in Fig 9A–9F, which demonstrate six time-stamped frames taken on February 15, March 15, April 15, May 15, June 15, and July 15, 2020. A proportional symbol map is used with semi-transparent red circles to represent the general MDRI. This visualization technique enhances clustering patterns, and there is a clear trend where the general medical burden shifted from the east coast of the U.S. to midwestern states. As of July 2020, it would seem that Louisiana, Mississippi, Georgia, Tennessee, Indiana, and Iowa are possibly suffering a new wave of medical resource deficiencies due to the rapid increase of accumulated active confirmed cases in some counties.

Furthermore, the spatiotemporal dynamics of local daily $MRDI_d$ is also illustrated in the dashboards. Since hospitalization data has been available only recently, we illustrate two frames taken on May 1, and August 1, 2020 (Fig 10A and 10B). The red circle symbols are semi-transparent, and county-level medical resource deficiencies are visually enhanced by searching the reddest clustering patterns in the map. During this COVID-19 infection period, it seems that Mississippi, Louisiana, Tennessee, and Indiana were suffering from medical resource deficiencies, which would have required special attention when relocating medical resources if necessary. These hotspots have been partially confirmed from local news reports. For example, there were 5,153 known presumptive cases with the total death toll of 201 in Mississippi on April 23, 2020 [28]; new cases of COVID-19 rose sharply on May 1 in East Baton

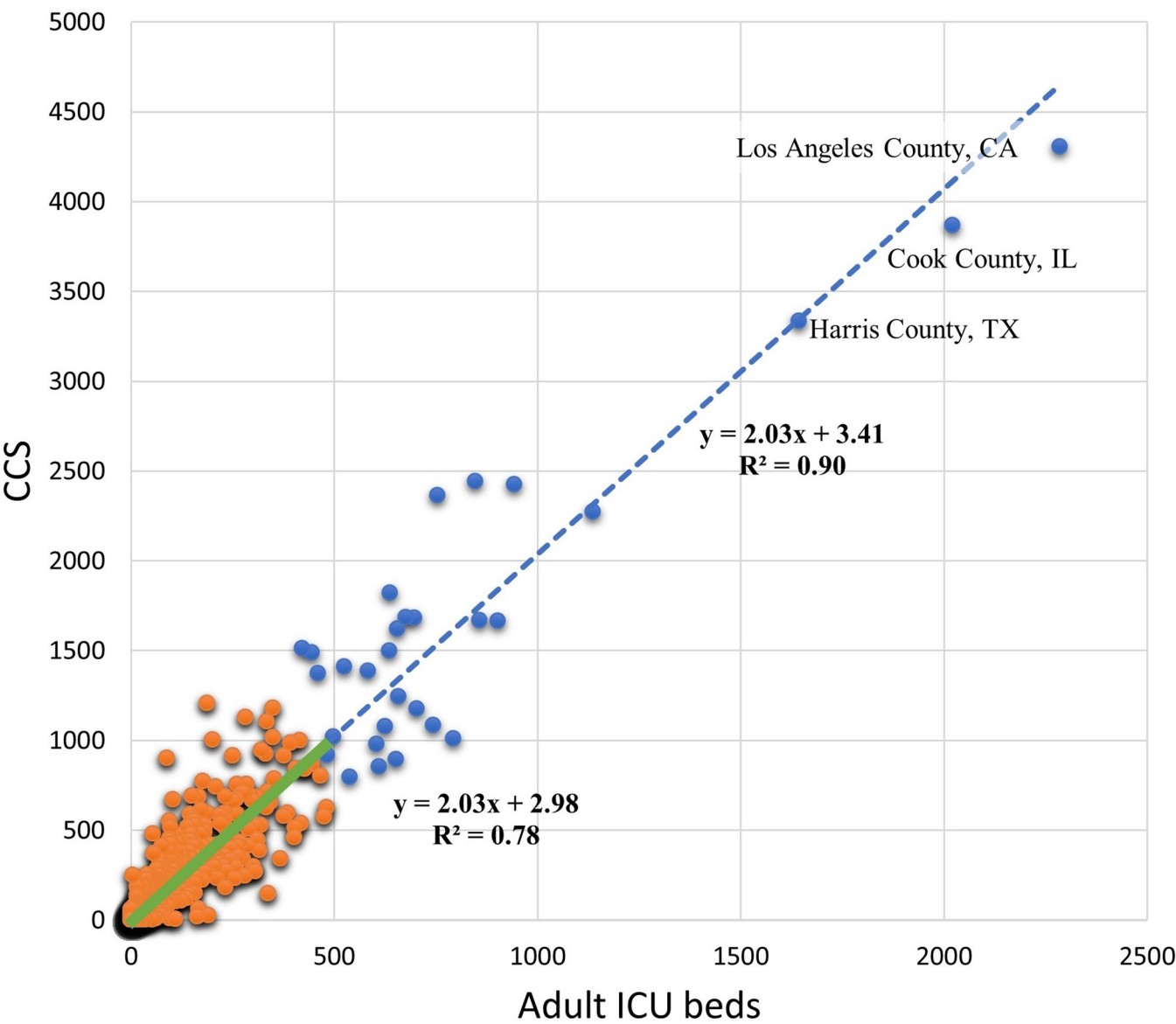

**Fig 7. The correlation between CCS and adult ICU beds at county level. Blue dots represent 30 counties with rich medical resources.** Orange dots represent the other 3113 counties. The blue dashed line is the overall regression line, and the green solid line is the regression line for the 3113 counties only.

Rouge, Louisiana, as deaths approached 350 in the region [29]; the nation's highest infection rate was in a county in Trousdale County, Tennessee, where 1,300 cases of Covid-19 were reported, and most of them traced back to a state correction center [30]; and Indiana passed 1,000 COVID-19 deaths on April 29, 2020 [31].

### 4.3. Spatiotemporal visualization and analysis interface

In the center of the dashboard, several map layers could be selected to show the general spatial distribution of MRDI, death rate, infection rate and active cases over licensed beds per capita. After interactive map scaling (by zooming in/out) and moving (by dragging) operations, or using the polygon selection tool, the charts and rank list are linked and self-adapted to the analysis region of interest to a user. By clicking the polygon of a selected county, attribute

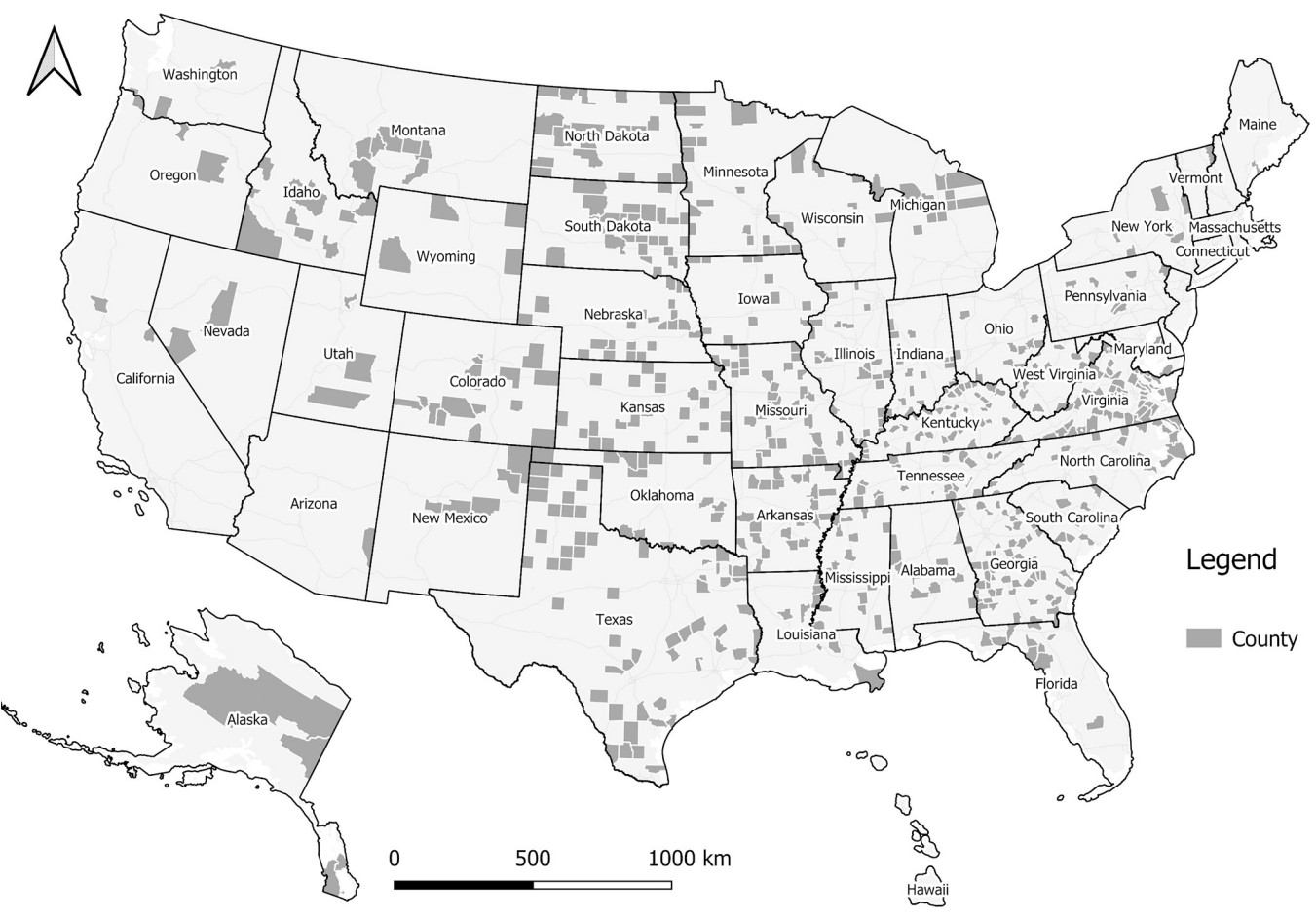

**Fig 8. The 671 counties without licensed beds or CCS.**

information about medical resources and COVID-19 related data would popup and the relevant chart is automatically updated in the dashboard.

Northern Tennessee State is presented as a use case to show the possible interactive analysis (Fig 4). Since western and east coast regions have more medical resources than central regions (Fig 6A), and the states along the Mississippi River in the southern U.S. show a high risk (Fig 10), we zoom in on the map and select the nearest region with the largest red bubble in Tennessee (Fig 4). Thirty counties are selected as a result, and relevant numbers are calculated and presented in dashboard charts. The medical bed pie chart shows ICU beds are 10.89% in overall licensed beds, and the medical staff pie chart shows the nurses group is the highest (55.77%) followed by physicians (44.01%), physician assistants (0.18%) and therapists (0.04%). The line chart shows a time-series trend for MRDI in the northern Tennessee area, and we find the index varied greatly between April 30, 2020 to May 1, 2020, which could be explained by the possible tracing of the virus to a correction center outbreak in Trousdale County [32]. On the right column of the dashboard, the risk factors of medical resource and infection rate is ranked by the selected region. Trousdale, Davidson, and Sumner County are the top 3 with highest infection risks, while Trousdale also shows the highest medical resource risk in this region. The case study in Fig 4 demonstrates the potential of our developed dashboard for interactive and visual analysis of specific regions of interest for policy makers, other stakeholders, and the general public.

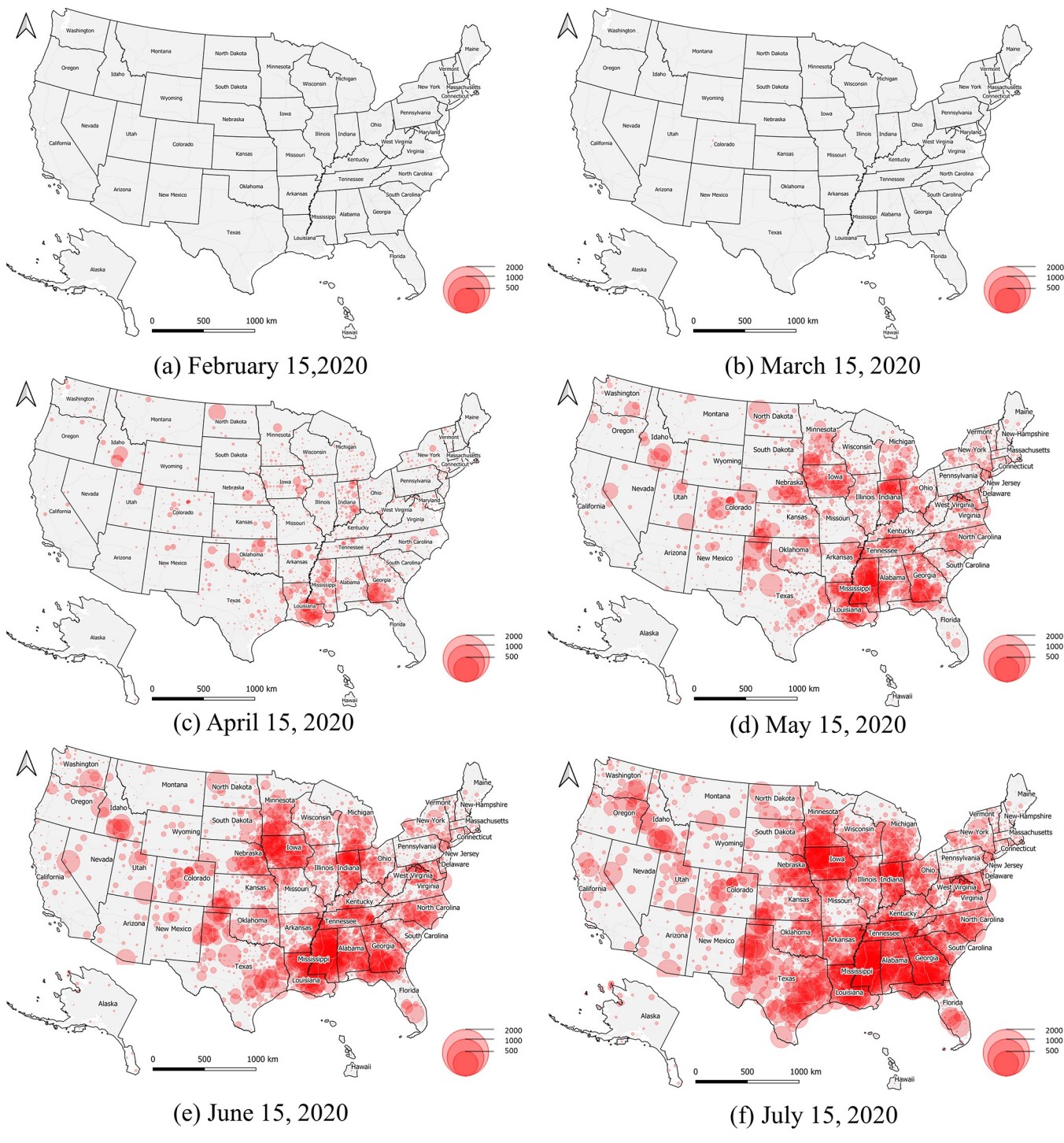

**Fig 9. General MRDI trend.**

## 5. Discussion and conclusions

In this study, a data-driven approach has been used to estimate the medical resource deficiencies or medical burden at county level during the COVID-19 pandemic across the U.S. Specifically, spatiotemporal data analysis methods including feature extraction, database structured

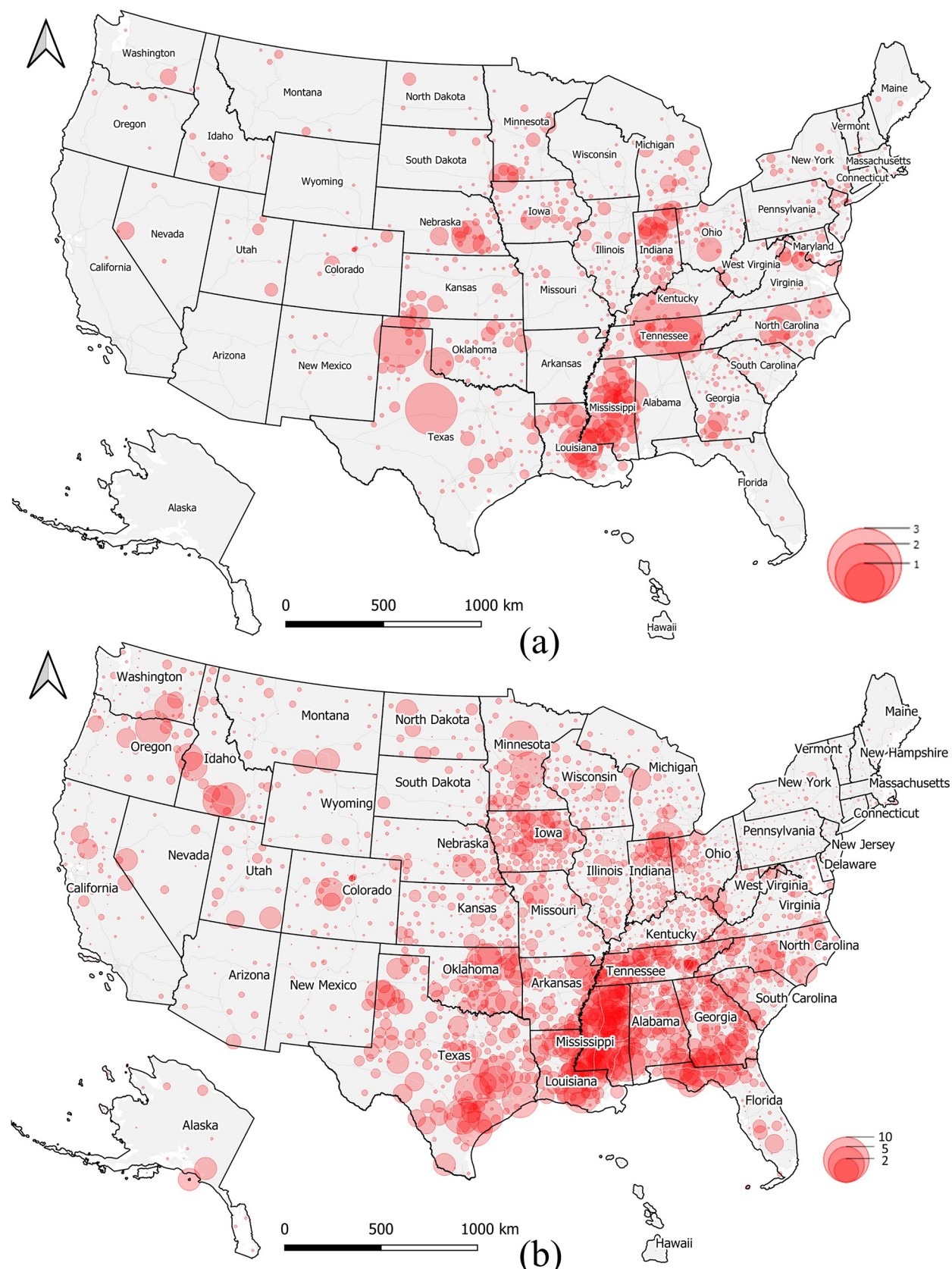

**Fig 10.** Daily medical burden $MRDI_d$ trend on May 1 (a) and August 1, 2020 (b).

query (SQL), data fusion or aggregation, linear regression analysis, and spatial statistics were used to extract medical resource features and patient statistics, such as hospital beds, CCS, local population, COVID-19 confirmed case numbers, and hospitalization data at county level. The average length of stay (ALOS) was then estimated from hospitalization data at state level, and the hospitalized rate were calculated at state and county level. Based on these datasets, we developed two medical resource deficiency indices MRDI and $MRDI_d$ that measure the local medical burden from two different perspectives. The first index represents the number of accumulated active confirmed cases normalized by local maximum potential medical resources; and the second one represents the number of hospitalized patients that can be supported per ICU beds per critical care staff. The related medical resource data, MRDI and $MRDI_d$ were visualized and analyzed using a dynamic spatiotemporal platform created through ArcGIS Pro Dashboards, which is a convenient way to enhance the clustering patterns and trends.

Our analysis showed that (1) the spatial distribution of medical resources (hospital beds, ICU beds, and CCS) at county level is highly heterogeneous across the U.S., and ICU beds and CCS are not spatially highly correlated; (2) MRDI and $MRDI_d$ can provide new insights into the U.S. pandemic preparedness and local dynamics relating to medical burdens during a peak period in the COVID-19 pandemic; and (3) a data-driven dynamic spatiotemporal framework is a powerful data visualization tool to illustrate the trends of MRDI / $MRDI_d$ and other medical-related statistics.

It is worth noting that we have not considered the number of discharged patients due to lack of data, leading to a possible slight underestimate of ALOS during the COVID-19 rapid infection period. As a result, $MRDI_d$ may also be slightly underestimated. We also did not consider the ratio of ICU patients and acute hospitalized patients due to lack of data, and assumed all hospitalized patients were treated as ICU cases. As a result, $MRDI_d$ was possibly overestimated, and the values calculated here should be viewed as the upper limit of local medical burdens. Some other uncertainties include (1) the numbers of registered hospital beds and CCS could be incomplete or not up-to-date, although the most recent Definitive Healthcare and NPI databases have been used, so the medical resources could be underestimated, (2) critically ill patients in counties without ICU beds and CCS would be sent to neighboring counties with sufficient medical resources, (3) some numbers of experienced ICU staff may become ill, (4) the number of trained professionals may have increased based on emergent recruiting, and (5) the capacity in ICUs and emergency rooms may have been expanded during the crisis. However, $MRDI_d$ can still serve as a useful indicator to measure the county-level medical resource deficiencies, and this index can be improved once more public health data are available in the future. Furthermore, it could provide reasonable evidence for policy makers in local and state governments to assess their medical inventories and staff resources, and provide preparedness for decision of re-opening the economies and public life.

In the future, our work can be combined with epidemic models to either provide driving parameters or calibrate the models and predict the local medical burdens. The spatiotemporal analysis used in this study can be extended to include remote sensing data, social media data, and mobile traffic flow data to estimate severity of pandemic or predict the outbreak cases in the U.S. and other counties.

## Author Contributions

**Conceptualization:** Dexuan Sha, Xin Miao, Yuyang Tian, Chaowei Yang.

**Data curation:** Dexuan Sha, Hai Lan, Shiyang Ruan, Yifei Tian, Chaowei Yang.

**Formal analysis:** Dexuan Sha, Xin Miao, Hai Lan, Shiyang Ruan, Yifei Tian.

**Funding acquisition:** Xin Miao, Chaowei Yang.

**Investigation:** Yuyang Tian, Chaowei Yang.

**Methodology:** Dexuan Sha, Xin Miao, Hai Lan, Yifei Tian, Yuyang Tian, Chaowei Yang.

**Project administration:** Xin Miao, Chaowei Yang.

**Resources:** Yuyang Tian.

**Software:** Dexuan Sha, Hai Lan, Yifei Tian.

**Supervision:** Xin Miao, Chaowei Yang.

**Validation:** Dexuan Sha, Xin Miao, Hai Lan, Shiyang Ruan, Yuyang Tian, Chaowei Yang.

**Visualization:** Dexuan Sha, Hai Lan.

**Writing – original draft:** Dexuan Sha, Xin Miao, Hai Lan, Kathleen Stewart, Yuyang Tian, Chaowei Yang.

**Writing – review & editing:** Kathleen Stewart, Chaowei Yang.

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
