## [Decision Letter · Decision Letter 0]

10 Aug 2020

PONE-D-20-14757

Spatiotemporal Analysis of Medical Resource Deficiencies in the U.S. under COVID-19 Pandemic

PLOS ONE

Dear Dr. Miao,

Thank you for submitting your manuscript to PLOS ONE. After careful consideration, we feel that it has merit but does not fully meet PLOS ONE’s publication criteria as it currently stands. Therefore, we invite you to submit a revised version of the manuscript that addresses the points raised during the review process.

We look forward to receiving your revised manuscript.

Kind regards,

Wenbin Tan

Academic Editor

PLOS ONE

2.We note that [Figure(s) 3, 4, 5, 6, 8, 9 and 10] in your submission contain [map/satellite] images which may be copyrighted. All PLOS content is published under the Creative Commons Attribution License (CC BY 4.0), which means that the manuscript, images, and Supporting Information files will be freely available online, and any third party is permitted to access, download, copy, distribute, and use these materials in any way, even commercially, with proper attribution. For these reasons, we cannot publish previously copyrighted maps or satellite images created using proprietary data, such as Google software (Google Maps, Street View, and Earth). For more information, see our copyright guidelines: http://journals.plos.org/plosone/s/licenses-and-copyright.

1.    You may seek permission from the original copyright holder of Figure(s) [ 3, 4, 5, 6, 8, 9 and 10] to publish the content specifically under the CC BY 4.0 license. 

Additional Editor Comments:

Do not need to change the article format for your revision.

Reviewers' comments:

Reviewer #1: An estimation of medical resource deficiencies or medical burden at county level is developed using data-driven approach. The developed approach is performed during the COVID-19 pandemic from February 15, 2020 to May 1, 2020 in the U.S. Multiple data sources were used to extract local population, hospital beds, critical care staff, COVID-19 confirmed case numbers, and hospitalization data at county level. The average length of stay from hospitalization data at state level is estimated, and the hospitalized rate at both state and county level are calculated. Then, two medical resource deficiency indices that measure the local medical burden are developed based on the number of accumulated active confirmed cases normalized by local maximum potential medical resources, and the number of hospitalized patients that can be supported per ICU beds per critical care staff, respectively. The medical resources data, and the two medical resource deficiency indices are illustrated in a dynamic spatiotemporal visualization platform based on ArcGIS Pro Dashboards.

The manuscript is well written and structured; however, there are some comments which as follows :

1 – In page 6 – line 108 , it is recommended to move the URL to the references .

2 – in page 14 – line 262 , line 267, and line 268, “ MRDI_d” should be in math mode .

3 – in page 15 – line 270 , “ MRDI_d” should be in math mode .

4 – in page 19 – line 349 , it is recommended to move the URL the references .

5 – in page 25 – line 476 , try to replace reference [10] by another reliable source since the arXiv articles are not peer-reviewed .

6 – in page 27 – line 505 , try to replace reference [23] by another reliable source since the arXiv articles are not peer-reviewed .

7 – in the reference, it is recommended to check the journal guide for authors on how to write URL in the references .

Reviewer #2: The article provide very useful analysis of the health infrastructure in US settings and highlights the capability and quality in terms of the medical equipment and healthcare staff in managing a pandemic such as Covid-19. Although the inferences drawn from the analysis are relevant and important the manuscript can be accepted as short communication or a editorial. The manuscript has limitations in terms of study design and the information presented to be considered as a original research article for publication. It is more like a audit in its present state.

Reviewer #3: Abstract: The manuscript is technically sound and the data generated through synthesis supported the conclusion. However, brief background information was lacked in the very beginning of the abstract. Moreover, the key words would have been placed at the end of the abstract.

Introduction: The introduction should have been started with a precise statement explaining about pandemic and COVID-19.

Methods: No single figure was presented in the manuscript. I regarded this as a part of the plan by chief and office based editor to minimize bulkiness of the manuscript. The resource-medical interventions compatibility analyses have been performed appropriately and rigorously. All relevant data were included in the manuscript.

Results: Nicely presented and interpreted

Discussion and Conclusions: The findings were discussed and compared. Conclusions were data based.

Reviewer Summary: Minor language edition was made by track changes to bring the manuscript to the level of high standard. All other comments were included by track changes within the manuscript.

Reviewer #4: The authors have made a commendable effort in analyzing the medical burden in U.S. on a state level. The graphs and maps included in the manuscript provide a perfect illustration of the conclusions drawn in the study. This study highlights the medical resources deficiency and can help prepare the ground for the future preparedness for the next medical emergency of the century.

Reviewer #5: The author confined the analysis to four months i.e. from February 2020 to May 2020 whereas COVID 19 get peak in July 2020 in USA. So result about preparedness cannot be realistic without incorporating the peak data.

Discharged patient number should not be consider zero. An estimated value can be consider on the basis of available data to optimize the results.

---

## [Author Response · Author response to Decision Letter 0]

18 Sep 2020

A rebuttal letter that responds to each point raised by the academic editor and reviewers is submitted with the revised manuscript (Response to Reviewers.docx).

---

## [Editor Report · Decision Letter 1]

25 Sep 2020

Spatiotemporal Analysis of Medical Resource Deficiencies in the U.S. under COVID-19 Pandemic

PONE-D-20-14757R1

Dear Dr. Miao,

We’re pleased to inform you that your manuscript has been judged scientifically suitable for publication and will be formally accepted for publication once it meets all outstanding technical requirements.

Kind regards,

Wenbin Tan

Academic Editor

PLOS ONE

Reviewers' comments: The revised manuscript has addressed the comments from reviewers well. It is a very important and timely paper to analyze the medical resource deficiencies in US under this pandemic.

---

## [Editor Report · Acceptance letter]

5 Oct 2020

PONE-D-20-14757R1 

Spatiotemporal Analysis of Medical Resource Deficiencies in the U.S. under COVID-19 Pandemic 

Dear Dr. Miao:

I'm pleased to inform you that your manuscript has been deemed suitable for publication in PLOS ONE. Congratulations! Your manuscript is now with our production department. 

Kind regards, 

on behalf of

Dr. Wenbin Tan 

Academic Editor

PLOS ONE